# Asymmetric three-component olefin dicarbofunctionalization enabled by photoredox and copper dual catalysis

Peng-Zi Wang[1,3], Yuan Gao[1,3], Jun Chen[1], Xiao-Die Huan[1], Wen-Jing Xiao [1,2✉] & Jia-Rong Chen [1✉]

The intermolecular three-component alkene vicinal dicarbofunctionalization (DCF) reaction allows installation of two different carbon fragments. Despite extensive investigation into its ionic chemistry, the enantioseletive radical-mediated versions of DCF reactions remain largely unexplored. Herein, we report an intermolecular, enantioselective three-component radical vicinal dicarbofunctionalization reaction of olefins enabled by merger of radical addition and cross-coupling using photoredox and copper dual catalysis. Key to the success of this protocol relies on chemoselective addition of acyl and cyanoalkyl radicals, generated in situ from the redox-active oxime esters by a photocatalytic N-centered iminyl radical-triggered C-C bond cleavage event, onto the alkenes to form new carbon radicals. Single electron metalation of such newly formed carbon radicals to TMSCN-derived **L1**Cu(II)(CN)$_2$ complex leads to asymmetric cross-coupling. This three-component process proceeds under mild conditions, and tolerates a diverse range of functionalities and synthetic handles, leading to valuable optically active $\beta$–cyano ketones and alkyldinitriles, respectively, in a highly enantioselective manner (>60 examples, up to 97% ee).

---

[1] CCNU-uOttawa Joint Research Centre, Key Laboratory of Pesticides & Chemical Biology Ministry of Education, College of Chemistry, Central China Normal University, Wuhan, Hubei, China. [2] State Key Laboratory of Applied Organic Chemistry, Lanzhou University, Lanzhou, China. [3] These authors contributed equally: Peng-Zi Wang, Yuan Gao. ✉email: wxiao@mail.ccnu.edu.cn; chenjiarong@mail.ccnu.edu.cn

Alkenes are arguably one of the most privileged and versatile chemicals in organic synthesis because a diverse range of functional groups could be readily introduced across the C=C π system by many classic vicinal difunctionalization reactions[1–4]. In this context, one of the most investigated and fundamental transformations is the intermolecular three-component alkene vicinal dicarbofunctionalization (DCF) reaction, which allows installation of two different carbon fragments. While these reactions have been extensively explored in ionic chemistry, radical mediated, especially the enantioseletive versions of such reaction class remain largely unexplored[5–10]. Given the unique reactivity modes of radical species, the development of radical-mediated alkene vicinal DCF reactions would enable a wider variety of functional groups and synthetic handles to be incorporated, and hence produce valuable target molecules[11–15]. While particularly promising, controlling the stereoselectivity in radical alkene vicinal DCF reactions is a long-standing and fundamental challenge due to the intrinsically high reactivity and instability of radical intermediates[16–18]. In recent years, owing to their unique single-electron transfer (SET) ability and good coordination with chiral ligands, chiral copper catalysis opened a new and robust platform for the development of asymmetric radical-mediated alkene vicinal DCF reactions (Fig. 1a)[19–21]. For example, the Liu group disclosed that carbon-centered radicals, which are formed in situ by decarboxylative C–C bond cleavage of N-hydroxy-phthalimide esters[22], or upon addition of CF₃ and (fluoro)alkyl[23–28] or aryl radicals[29] to alkenes or enamides, could

couple with TMSCN, boronic acid, or alkyne-derived chiral copper(II) complexes, leading to alkene vicinal DCF products with excellent enantiomeric excess. Recently, Zhang[30] and Liu[31–33] reported elegant examples of copper-catalyzed highly enantioselective radical alkene 1,2-DCF with diaryliodonium salts and alkyl halides as radical sources, respectively. Despite the broad synthetic applicability of these methods, however, the intrinsic redox potential window of copper catalysts, which is critical to the generation of radicals, still results in significant limitations on the scope of radical precursors.

In recent years, visible-light photoredox catalysis has emerged as a powerful tool for organic chemists to develop many elusive radical-mediated chemical transformations with high levels of functional group tolerance[34–37]. This activation mode also provides a promising approach for the development of radical multicomponent reactions (in some cases with excellent stereoselectivity)[38–43]. For example, the Studer group recently developed an asymmetric three-component cascade reaction of quinolines or pyridines with enamides using α-bromo carbonyl compounds as radical precursors under photoredox and phosphoric acid dual catalysis[44]. Using this strategy, a range of chiral γ-amino acid derivatives could be achieved with high chemo-, regio-, and enantioselectivity. Recently, our group[45–48] and others[49–55] introduced readily accessible redox-active oxime derivatives as precursors to generate iminyl radicals under SET reduction or oxidation conditions (Fig. 1b). The resultant iminyl radicals further triggered smooth formation of sp³ cyanoalkyl and

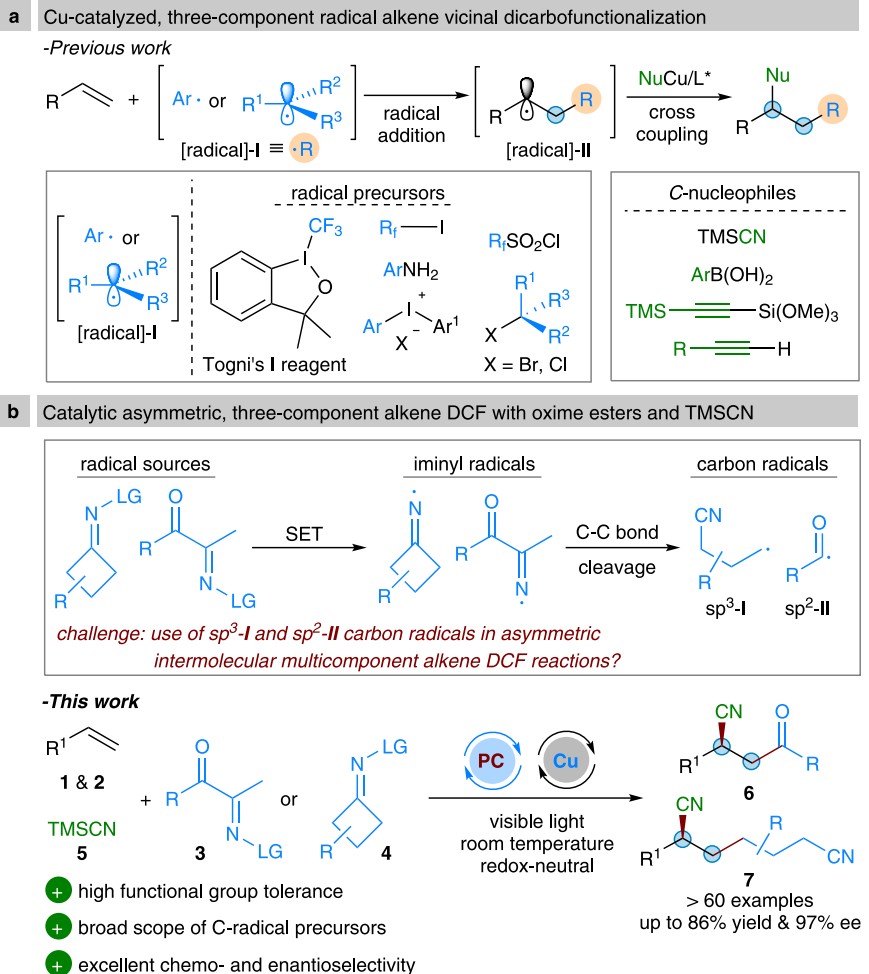

**Fig. 1 Catalytic asymmetric three-component radical dicarbofunctionalization reactions of alkenes. a** Cu-catalyzed, three-component radical alkene vicinal dicarbofunctionalization. **b** Catalytic asymmetric three-component alkene DCF with oxime esters and TMSCN.

sp² acyl[56–58] radicals via a C–C bond cleavage process. Despite extensive synthetic utility of these carbon radicals, to our knowledge, their engagement in asymmetric multicomponent alkene vicinal DCF reactions is still challenging and unexplored[59,60]. As a result, we aimed at developing a catalytic asymmetric three-component radical vicinal DCF reaction of alkenes with oximes and TMSCN (Fig. 1b). This protocol would provide an efficient and general approach for preparation of valuable optically active β-cyano ketones and alkyldinitriles[61–63].

## Results

**Optimization of reaction conditions**. Our optimization began by reacting 2-vinylnaphthalene **1a** with oxime ester **3a** and TMSCN in a ratio of 1:3:3 in DMA under the dual photoredox and copper catalysis (Table 1). After some experimentation (see supporting information for more details), we found that the target three-component reaction indeed occurred to give a moderate yield of desired product **6aa** with 88% ee, when using photocatalyst *fac*-Ir(ppy)₃ (1 mol%), and a combination of Cu(CH₃CN)₄PF₆ (0.5 mol%) and Box-type ligand **L1** (0.6 mol%) under irradiation of purple LEDs (Table 1, entry 1). However, several competing processes were also involved. For example, in addition to **6aa**, a significant amount of side products **sp-1**, **sp-2**, and **sp-3** were also detected, which might result from acyl radical **3a-II**-mediated two-component cross-coupling with **1a** and

TMSCN, or its own dimerization. A brief screening of typical solvents such as DMF, CH₃CN, and THF showed that DMA was still the best of choice in terms of reaction efficiency (Table 1, entry 1 vs entries 2–5). Notably, when the catalyst loading of *fac*-Ir(ppy)₃ was decreased to 0.8 mol%, a cleaner reaction was observed, and a 64% yield of **6aa** was obtained without effect on the enantioselectivity (Table 1, entry 6). An extensive survey of other commonly used copper salts and chiral ligands established that the combination of Cu(CH₃CN)₄PF₆ and Box-type ligand **L1** were superior to others (Table 1, entries 6–8). Further optimization studies with respect to the loading of copper salt and concentration confirmed that a combination of 1.5 mol% of Cu(CH₃CN)₄PF₆ and 2.25 mol% of ligand **L1** with 0.8 mol% of *fac*-Ir(ppy)₃ at a concentration of 0.04 M gave the best results, with **6aa** being isolated with 74% yield and 90% ee (Table 1, entry 11). It was postulated that rational tuning of catalyst loading can help regulate the concentration of the reactive radical species, thus suppressing the competing side reaction pathways. Then, we examined other commonly used light sources including blue LEDs (2 × 3 W, λmax = 460 nm) and CFL lamp (40 W) in the model reaction (entries 12 and 13). Both reactions could also work to give the desired products with 90% ee, but in only moderate yields due to decreased conversion and formation of considerable amounts of byproducts. These results showed that the wavelength of the light has notable effect on the reaction efficiency. As expected, a series of control experimental results

---

**Table 1 Optimization of the reaction conditions.**

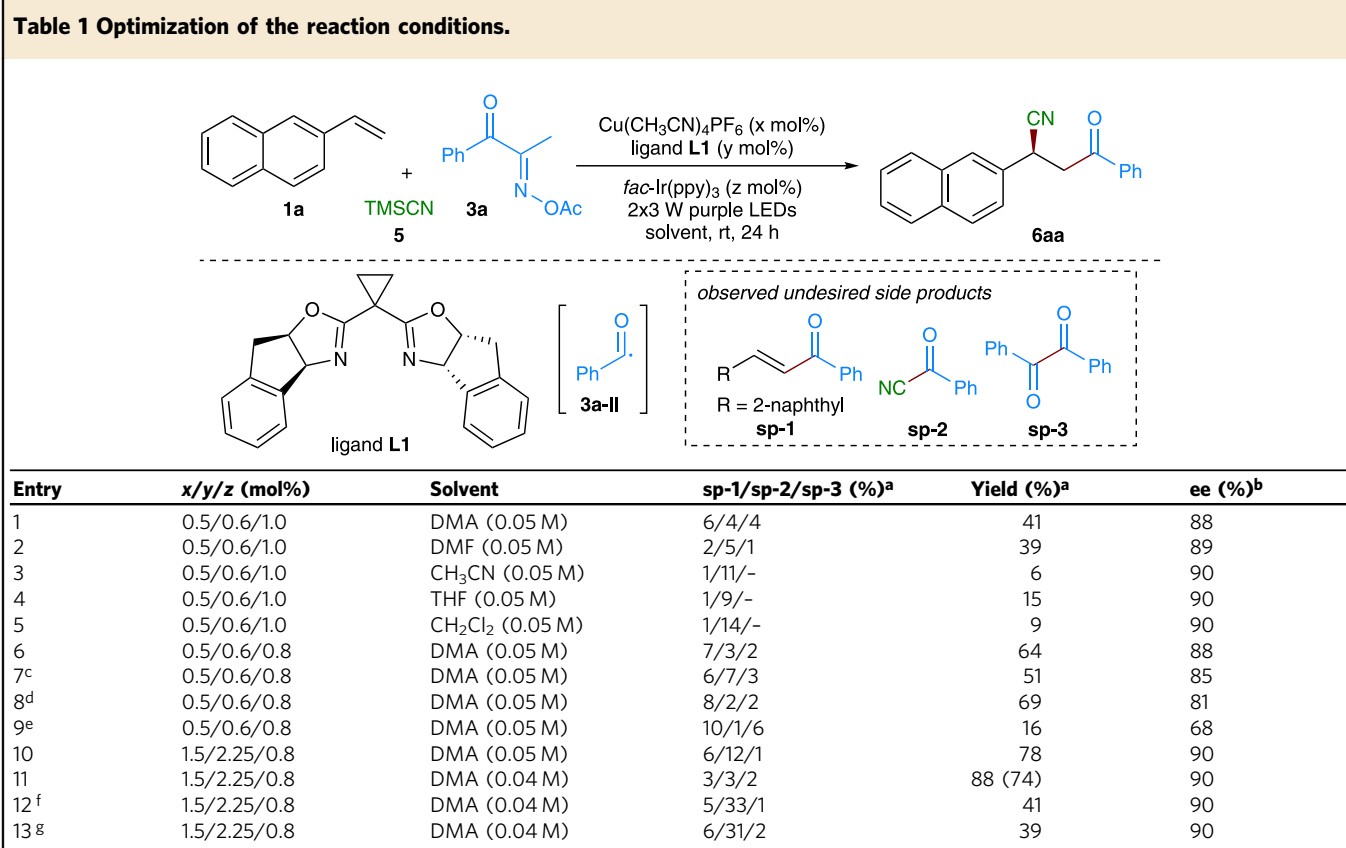

| Entry | x/y/z (mol%) | Solvent | sp-1/sp-2/sp-3 (%)[a] | Yield (%)[a] | ee (%)[b] |
|---|---|---|---|---|---|
| 1 | 0.5/0.6/1.0 | DMA (0.05 M) | 6/4/4 | 41 | 88 |
| 2 | 0.5/0.6/1.0 | DMF (0.05 M) | 2/5/1 | 39 | 89 |
| 3 | 0.5/0.6/1.0 | CH₃CN (0.05 M) | 1/11/– | 6 | 90 |
| 4 | 0.5/0.6/1.0 | THF (0.05 M) | 1/9/– | 15 | 90 |
| 5 | 0.5/0.6/1.0 | CH₂Cl₂ (0.05 M) | 1/14/– | 9 | 90 |
| 6 | 0.5/0.6/0.8 | DMA (0.05 M) | 7/3/2 | 64 | 88 |
| 7[c] | 0.5/0.6/0.8 | DMA (0.05 M) | 6/7/3 | 51 | 85 |
| 8[d] | 0.5/0.6/0.8 | DMA (0.05 M) | 8/2/2 | 69 | 81 |
| 9[e] | 0.5/0.6/0.8 | DMA (0.05 M) | 10/1/6 | 16 | 68 |
| 10 | 1.5/2.25/0.8 | DMA (0.05 M) | 6/12/1 | 78 | 90 |
| 11 | 1.5/2.25/0.8 | DMA (0.04 M) | 3/3/2 | 88 (74) | 90 |
| 12[f] | 1.5/2.25/0.8 | DMA (0.04 M) | 5/33/1 | 41 | 90 |
| 13[g] | 1.5/2.25/0.8 | DMA (0.04 M) | 6/31/2 | 39 | 90 |

Reaction conditions: **1a** (0.1 mmol), **3a** (0.3 mmol), TMSCN (0.3 mmol), Cu(CH₃CN)₄PF₆ (x mol%), ligand **L1** (y mol%), *fac*-Ir(ppy)₃ (z mol%), solvent (2.0–2.5 mL), 2 × 3 W purple LEDs, at room temperature.
CFL compact fluorescent lamp, ppy 2-phenylpyridine, DMA N,N-dimethylacetamide, DMF N,N-dimethylformamide.
[a]Yields were determined by GC analysis, with isolated yield in parentheses.
[b]Determined by HPLC analysis on a chiral stationary phase.
[c]With CuCl.
[d]With CuI.
[e]With Cu(OTf)₂.
[f]Under the irradiation of 2 × 3 W blue LEDs (λmax = 460 nm) at room temperature.
[g]Under the irradiation of CFL (40 W) at room temperature.

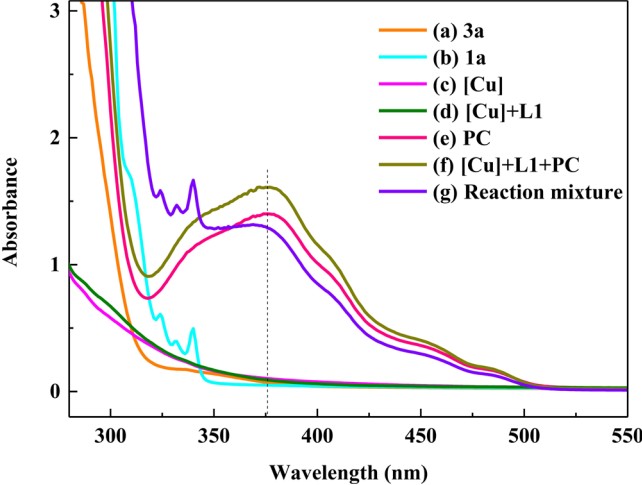

**Fig. 2 Absorption spectra. a** DMA solution of **3a**. **b** DMA solution of **1a**. **c** DMA solution of Cu(CH$_3$CN)$_4$PF$_6$. **d** DMA solution of Cu(CH$_3$CN)$_4$PF$_6$/**L1**. **e** DMA solution of *fac*-Ir(ppy)$_3$. **f** DMA solution of Cu(CH$_3$CN)$_4$PF$_6$/**L1**/*fac*-Ir(ppy)$_3$. **g** Reaction mixture.

established that each component (light, photocatalyst, and copper salt) is critical to this asymmetric alkene vicinal DCF reaction (see supporting information for more details).

Then, we obtained UV–Vis spectra of the solution containing **3a**, **1a**, Cu(CH$_3$CN)$_4$PF$_6$, and photocatalyst alone, equimolar mixtures of Cu(CH$_3$CN)$_4$PF$_6$, **L1**, and photocatalyst, as well as the reaction mixture (Fig. 2). It was found that **3a**, **1a**, Cu (CH$_3$CN)$_4$PF$_6$, and Cu(CH$_3$CN)$_4$PF$_6$/**L1** did not show any very strong absorption bands around the visible region. In contrast, the UV–Vis spectra of photocatalyst, mixture of Cu (CH$_3$CN)$_4$PF$_6$/**L1**/PC and reaction mixture have the same absorption band at 375 nm, which implied that photocatalyst *fac*-Ir(ppy)$_3$ could be more efficiently excited by purple LEDs ($\lambda$max = 390 nm) than by the blue LEDs ($\lambda$max = 460 nm).

**Substrate scope**. With optimized reaction conditions established, we first investigated the substrate scope of alkenes by reacting with **3a** and TMSCN (Fig. 3). Notably, most of these alkenes are inexpensive and commercially available feedstock chemicals. As shown in Fig. 3a, aside from **1a**, simple neutral styrene **1b** and a range of styrene derivatives **1c–j** with electron-donating (e.g., Me, *t*Bu, and Ph) or electron-withdrawing (e.g., F, Cl, Br, OAc, and Bpin) functional groups at the *para*-position of the aromatic ring are well tolerated, furnishing the corresponding products **6ba–ja** in 64–79% yields with 86–90% ee. Notably, halide substituents, F, Cl, and Br, as well as Bpin moiety remained intact after the reaction, thereby facilitating further modifications at their positions (e.g., products **6fa–ha** and **6ja**). A 1.0 mmol scale reaction of **1b** also proceeded smoothly to give comparable results (**6ba**, 74% yield, 90% ee), demonstrating the scalability of this process. Moreover, the reactions with alkenes **1k–o** bearing common substituents, such as methoxy, methyl, fluoro, and bromo at the *meta*- or *ortho*-positions also worked well; and the expected products **6ka–oa** were isolated in 61–81% yields with 89–90% ee. 2-Vinylnaphthalene **1p** having a methoxy group and heterocycle-containing alkenes **1q–s** all proved to be suitable coupling partners, leading to **6pa–sa** with good yields and 86–93% ee. Remarkably, this protocol can also be successfully extended to biologically relevant molecule and pharmaceutical-derived styrene analogues (Fig. 3b). For instance, estrone, febuxostat-, and simple amino acid-derived alkenes **1t–v** reacted well to give the desired acylcyanation products **6ta**, **6ua**, and **6va** with good

stereoselectivity, respectively. As a result, our protocol should be of potential use for late-stage structural modification of drug and complex compounds. Unfortunately, the current catalytic system is not applicable to simple unactivated or electron-deficient alkenes.

Then, we continued to evaluate the generality of this asymmetric three-component reaction by using a representative set of oxime esters, which can be easily prepared in two steps from the relevant ketone precursors. As shown in Fig. 4a, a range of aryl ketone-derived oxime esters **3b–g** with electronically diverse functional groups (e.g., Me, OMe, *t*Bu, F, Cl, or Br) at the *para*-position of the phenyl ring reacted well with **1b** and TMSCN. And the expected alkene acylcyanation products **6bb–bg** were obtained with good yields (70–86%) and excellent enantiomeric excess (83–92% ee). As shown in the cases of **3h–k**, the change of the substitution pattern and steric hindrance of their phenyl ring has deleterious effect on the reaction efficiency or enantioselectivity, with the corresponding products **6bh–bk** being obtained with 67–85% yields and 89–92% ee. Single crystals of product **6bj** were obtained, and the absolute stereochemistry was determined to be *S* by X-ray crystallographic analysis (CCDC 2047031 (**6bj**) and 2047032 (**7ja**) contain the supplementary crystallographic data for this paper. These data can be obtained free of charge from The Cambridge Crystallographic Data Centre via www.ccdc.cam.ac.uk/data_request/cif), and all other coupling products were tentatively assigned by analogy with **6bj**. Remarkably, oxime esters **3m–p** derived from aliphatic ketone with various lengths of alkyl chains also reacted well with 2-vinylnaphthalene **1a** and TMSCN (Fig. 4b). The relative products **6am–ap** were isolated with 61–70% yields and 86–90% ee. We also preliminarily examined the cyclic oxime ester **3q** by reacting with **1a** and **5** with slightly increased amounts of catalysts. Unfortunately, the reaction proceeded slowly, giving only trace amount of the desired product **6aq** together with some byproducts (see supporting information for more details).

Encouraged by these results, we further attempted to extend the current dual photoredox and copper catalysis strategy to the asymmetric three-component vicinal DCF reaction of cycloketone-derived oxime esters, alkenes, and TMSCN (Fig. 5). Minor modification of the reaction conditions identified that a combination of organic photocatalyst Ph-PTZ (1.25 mol%) and Cu(CH$_3$CN)$_4$PF$_6$ (0.5 mol%)/ligand **L1** (0.6 mol%) enabled the desired reaction to proceed smoothly under irradiation of 2 × 3 W purple LEDs at room temperature (see supporting information for more details). This process also exhibited broad substrate scope and good functional tolerance with respect to both alkenes and oxime esters. As shown in Fig. 5a, a wide variety of commercially available styrenes containing neutral (**2a**), alkyl (**2b** and **2c**), electron-withdrawing (**2e–h**), or aryl (**2i–j**) groups at the *para*-position of the phenyl ring could react well with oxime ester **4a**. The corresponding dinitrile products **7aa–ja** were obtained in 51–75% yields with 83–96% ee. The absolute stereochemistry of **7ja** was also confirmed to be *S*-configuration by X-ray diffraction (CCDC 2047031 (**6bj**) and 2047032 (**7ja**) contain the supplementary crystallographic data for this paper. These data can be obtained free of charge from The Cambridge Crystallographic Data Centre via www.ccdc.cam.ac.uk/data_request/cif). Again, as shown in the reactions of alkenes **2k–n**, variation of the substitution pattern and steric hindrance of the aromatic ring could be well tolerated, leading to formation of products **7ka–na** with 51–78% yields and 82–97% ee. Moreover, the reactions of 2-vinylnaphthalene **2o** and substrates containing heterocycle-fused ring (**2p**) or heteroaryl groups (e.g., **2q**, **2r**) all proceeded well to give products **7oa–ra** with moderate to good yields and excellent enantioselectivity (84–90% ee). Notably, styrenes (e.g., **2s** and **2t**) derived from dihydroartemisinin and gibberellic acid could also

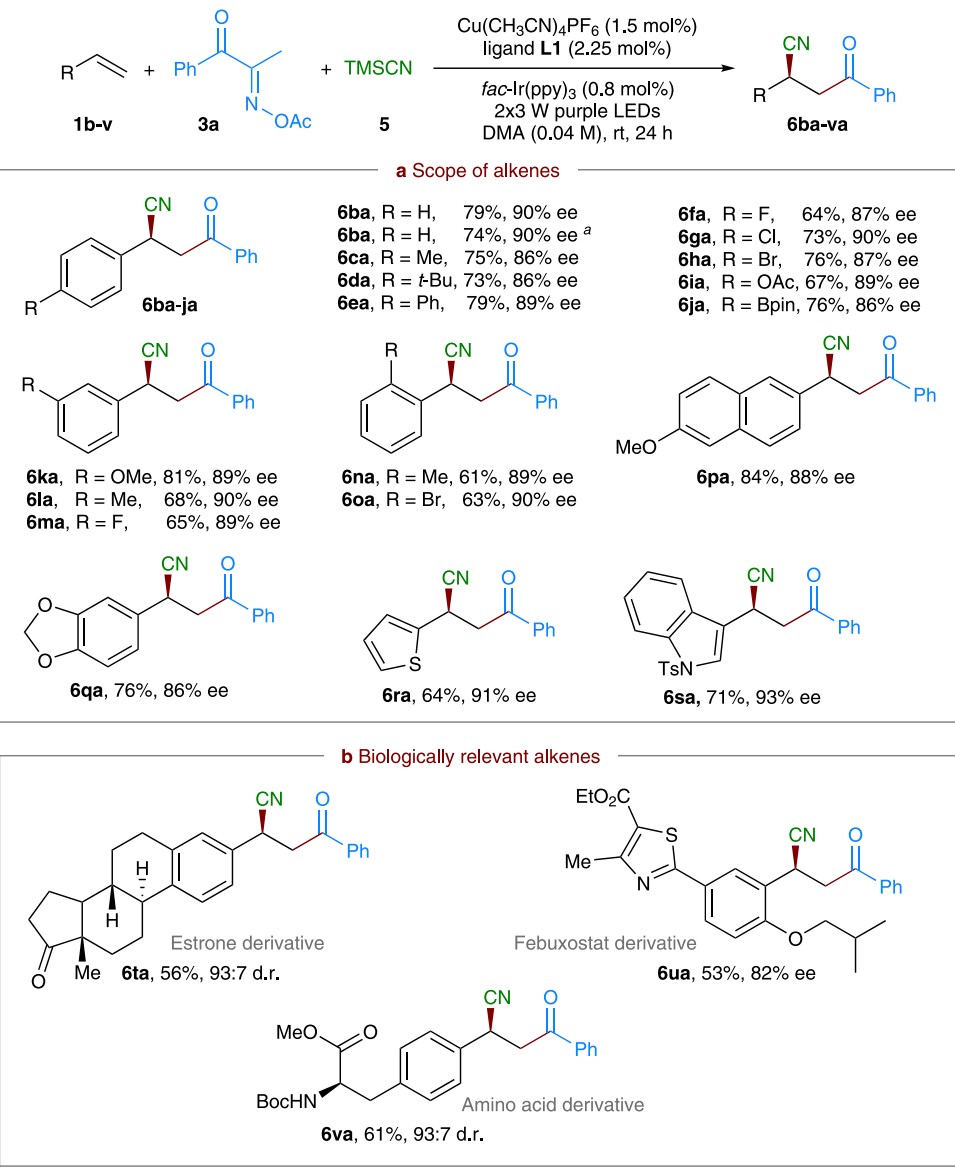

**Fig. 3 Scope of the alkenes in asymmetric three-component alkene acylcyanation.** Reaction conditions: **1** (0.1 mmol), **3a** (0.3 mmol), TMSCN (0.3 mmol), Cu(CH$_3$CN)$_4$PF$_6$ (1.5 mol%), ligand **L1** (2.25 mol%), *fac*-Ir(ppy)$_3$ (0.8 mol%), DMA (2.5 mL), 2 × 3 W purple LEDs, at room temperature. Isolated yields were reported. The ee and d.r. values were determined by HPLC analysis on a chiral stationary phase. [a]1.0 mmol scale reaction, 24 h. **a** Scope of alkenes. **b** Biologically relevant alkenes.

participate in the reaction with good stereoselectivity, suggesting that the method can potentially be used in the late-stage modification of pharmaceutically relevant compounds.

Finally, we turned our attention to study the substrate scope of cyclobutanone oxime esters by reacting with styrene **2i** and TMSCN (Fig. 5b). Both oxetan-3-one and 1-Cbz-3-azetidinone derived oxime esters **4b** and **4c** reacted well to afford **7ib** and **7ic** in moderate to good yields, with excellent enantioselectivity (87−89% ee). Monosubstituted oxime ester **4d** could participate in the reaction smoothly to deliver product **7id** as a mixture of diastereomers, with good yield and high enantioselectivity. Note that sterically demanding oxime esters **4e–g** also proved to be compatible with the reaction, giving the expected products **7ie–ig** in good yields with 85−92% ee.

**Synthetic applications**. To showcase the potential synthetic utility of this asymmetric method in the construction of valuable

skeletons, we performed diverse further transformations with the chiral β-cyano ketones and alkyldinitriles (Fig. 6)[64,65]. For example, the cyano group of **6ba** and **7ia** could be easily converted to amide group by Pd-catalyzed hydrolysis using stoichiometric acetaldoxime in refluxing aqueous EtOH, giving the corresponding products **8** and **9** with good yields, respectively (Fig. 6a). Moreover, treatment of **7ia** with NiCl$_2$/NaBH$_4$ and Boc$_2$O in MeOH allowed efficient sequential reduction and protection of both cyano groups, with aliphatic chiral amine **10** being obtained with 66% yield and 90% ee (Fig. 6b). The synthesis of chiral ester **11** can also be achieved by the treatment of **7ia** with alcoholysis. Notably, no notable loss of optical purity was detected in these manipulations.

**Mechanistic studies**. To gain some insight into the mechanism, we carried out several control experiments by using the substrates **1b**, **3a**, and TMSCN (Fig. 7). The target three-component

**Fig. 4 Scope of the oxime esters in asymmetric three-component alkene acylcyanation.** Reaction conditions: **1** (0.1 mmol), **3** (0.3 mmol), TMSCN (0.3 mmol), Cu(CH$_3$CN)$_4$PF$_6$ (1.5 mol%), ligand **L1** (2.25 mol%), fac-Ir(ppy)$_3$ (0.8 mol%), DMA (2.5 mL), 2 × 3 W purple LEDs, at room temperature. Isolated yields were reported. The ee and d.r. values were determined by HPLC analysis on a chiral stationary phase. **a** With alkene **1b**. **b** With alkene **1a**.

reaction was completely inhibited, when stoichiometric radical scavenger 2,2,6,6-tetramethyl-1-piperidinyloxy (TEMPO) was introduced (Fig. 7a). Instead, the relevant TEMPO-adduct **12** was obtained in 76% yield, suggesting the possible involvement of acyl radical **3a–I** in this process. Moreover, the reaction of radical clock substrate **13** having a cyclopropyl moiety also proceeded smoothly to give ring-opening product **14** in 64% yield with good stereoselectivity (Fig. 7b). These results indicated the intermediacy of radical species **13-A** and **13-B**, as well as the radical property of the reaction. Similar control experimental results were also observed in the case of cycloketone oxime ester-based asymmetric three-component reaction[63].

On the basis of these mechanistic studies and related literature[19–21,45–48], we proposed a dual photoredox and copper-catalyzed mechanism for the present asymmetric three-component reaction as depicted in Fig. 8. The reaction starts with SET reduction of redox-active oxime esters **3a** and **4a** by the excited state photocatalyst to give iminyl radicals **3a–I** and **4a–I**, with release of carboxylic anion (RCO$_2$$^-$). Then, **3a–I** and **4a–I** undergoes C–C bond β-cleavage to form acyl and cyanoalkyl radicals **3a-II** and **4a-II**. Further facile trap of these carbon radicals by styrene derivatives **1** and **2** forms relatively more stable benzylic radicals **III**. On the other hand, the initially formed carboxylic anion (RCO$_2$$^-$) could also facilitate the ligand exchange between **L1**/copper(I) complex and TMSCN to form **L1**Cu(I)CN species. Such **L1**Cu(I)CN complex can further be oxidized by the oxidizing photocatalyst (PC$^{•+}$) via a SET process, and undergoes another ligand exchange with TMSCN to form **L1**Cu(II)(CN)$_2$ complex,

regenerating ground-state photocatalyst to close the photoredox catalysis cycle. Finally, **L1**Cu(II)(CN)$_2$ traps the prochiral benzylic radical **III** to form a chiral high-valent Cu(III) complex **IV**, which undergoes the reductive elimination to afford the coupled product **6** or **7**, with regeneration of **L1**Cu(I)CN species to complete the copper catalysis cycle. It should be noted that an alternative process involving direct cyano transfer from the **L1**Cu(II)(CN)$_2$ complex to the benzylic radical **III** through an outer-sphere pathway is also possible. Notably, the whole process is redox neutral and does not need any external stoichiometric oxidants or reductants.

## Discussion

In summary, we have developed an intermolecular, highly enantioselective three-component radical vicinal DCF reaction of alkenes, using oxime esters and TMSCN, by dual photoredox and copper catalysis. Key to the success of this protocol relies on chemoselective addition of acyl and cyanoalkyl radicals, generated in situ from the redox-active oxime esters by a photocatalytic N-centered iminyl radical-triggered C–C bond cleavage event, onto the alkenes to form new carbon radicals. Single-electron metalation of such carbon radicals to TMSCN-derived **L1**Cu(II)(CN)$_2$ complex leads to asymmetric cross-coupling. This three-component reaction proceeds under mild conditions, and demonstrates broad substrate scope and high functional group tolerance, providing a general approach to optically active β–cyano ketones and alkyldinitriles. From a synthetic perspective, though the chiral products derived from

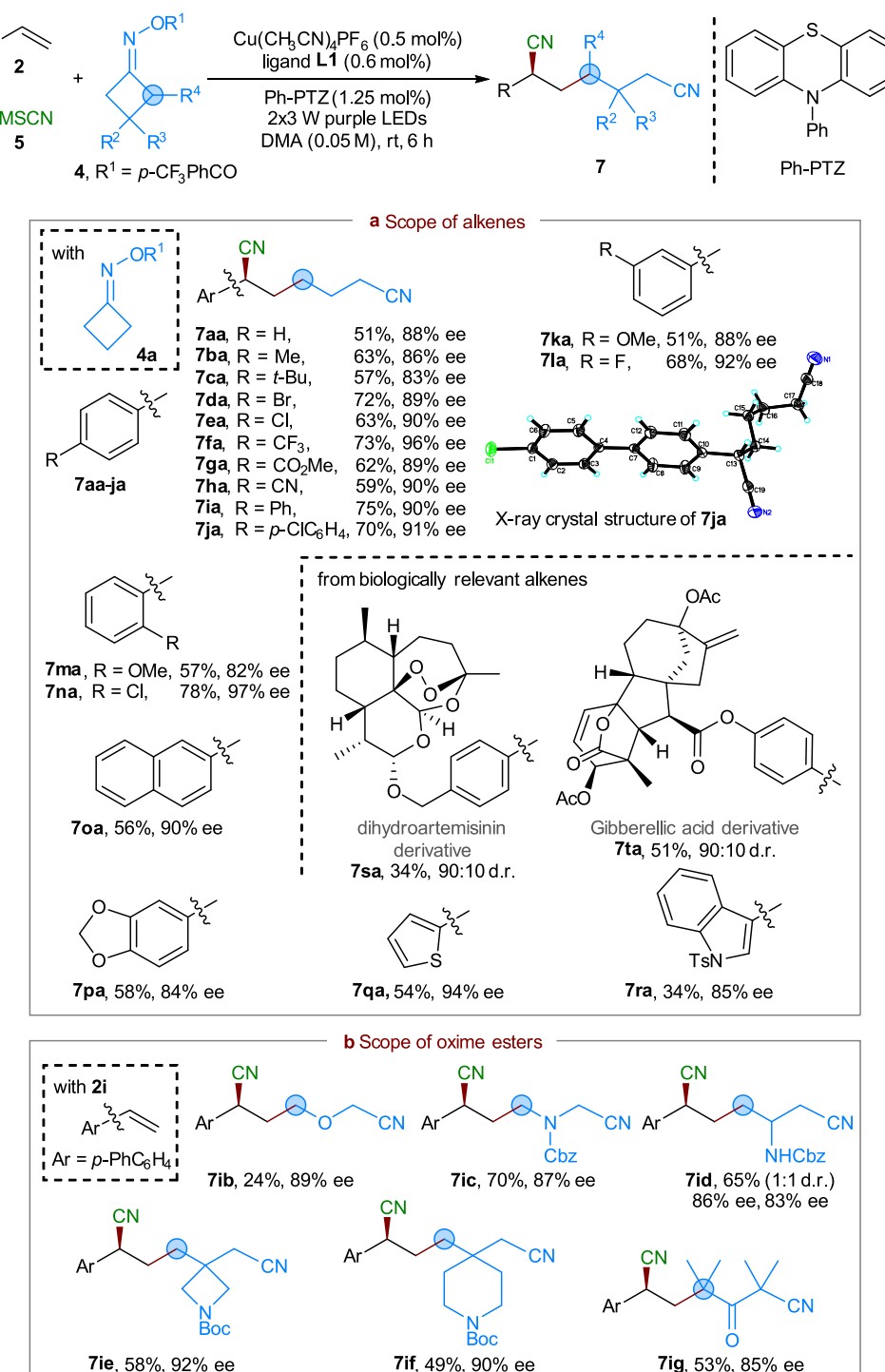

**Fig. 5 Scope of the alkenes and cycloketone oxime esters in asymmetric three-component alkene cyanoalkylcyanation reaction.** Reaction conditions: **2** (0.2 mmol), **4** (0.6 mmol), TMSCN (0.6 mmol), Cu(CH₃CN)₄PF₆ (0.5 mol%), ligand **L1** (0.6 mol%), Ph-PTZ (1.25 mol%), DMA (4.0 mL), 2 × 3 W purple LEDs, at room temperature. Isolated yields were reported. The ee and d.r. values were determined by HPLC analysis on a chiral stationary phase. **a** Scope of alkenes. **b** Scope of oxime esters.

acyclic ketone oxime esters can also be obtained by catalytic enantioselective conjugate addition strategies of cyanide to enones[66–68], our three-component protocol provides a modular and generally applicable approach for the synthesis of structurally diverse chiral β–cyano ketones. Moreover, the products derived from cyclic ketone oxime esters cannot be easily synthesized by the previous methods. Many exciting extensions of this strategy to other radical precursors and

nucleophiles can be envisaged; current investigations into these subjects are ongoing in our laboratory.

## Methods
**General procedure for the synthesis of products 6.** In a flame-dried 10 mL Schlenk tube equipped with a magnetic stirrer bar was charged sequentially with Cu (CH₃CN)₄PF₆ (0.56 mg, 0.0015 mmol) and chiral ligand **L1** (0.80 mg, 0.00225 mmol), followed by the addition of DMA (2.5 mL). Then the mixture was stirred at room

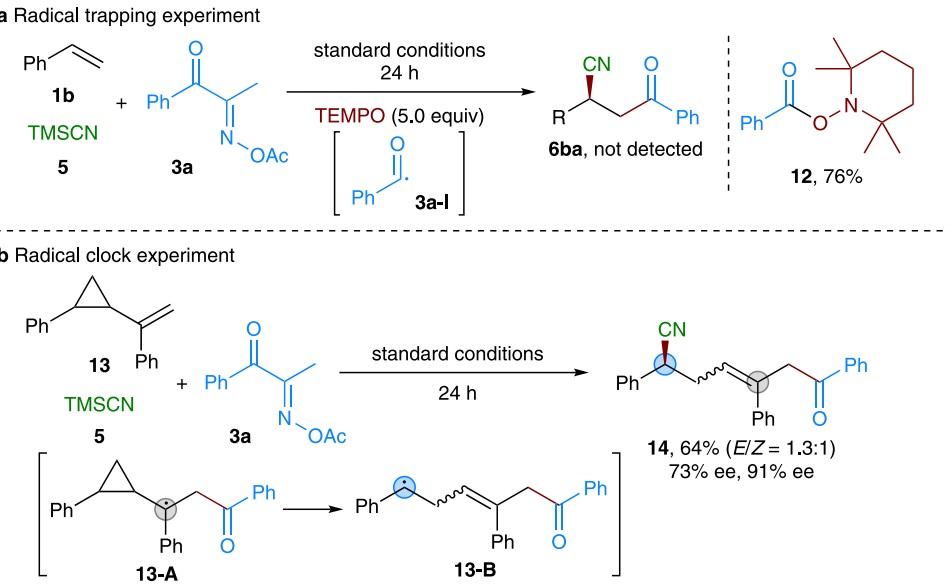

**Fig. 6 Synthetic applications. a** Pd-catalyzed hydrolysis of **6ba** and **7ia** for the synthesis of amide. **b** Reductive amination and alcoholysis of dinitriles **7ia**.

**Fig. 7 Mechanistic studies. a** Radical trapping experiment. **b** Radical clock experiment.

temperature for 30 min. To the resulting mixture were added **3** (0.30 mmol), **1** (0.10 mmol), and *fac*-Ir(ppy)₃ (0.53 mg, 0.0008 mmol). Then, the resulting mixture was degassed (three times) under argon atmosphere. After that, TMSCN (0.3 mmol) was added into the mixture. At last, the mixture was stirred at a distance of ~1 cm from a 2 × 3 W purple LEDs at room temperature for 24 h until the reaction was completed, as monitored by TLC analysis. The reaction mixture was quenched with water (10 mL), diluted with EtOAc (3 × 10 mL), washed with NaCl (aq.), and dried over with anhydrous Na₂SO₄. After filtration and concentration, the residue was purified by silica gel chromatography affording the final products.

**General procedure for the synthesis of products 7**. In a flame-dried 10 mL Schlenk tube equipped with a magnetic stirrer bar was charged sequentially with Cu(CH₃CN)₄PF₆ (0.001 mmol), chiral ligand **L1** (0.0012 mmol), and organo-photocatalyst Ph-PTZ (0.0025 mmol), followed by the addition of DMA (4 mL). Then the mixture was stirred at room temperature for 30 min. To the resulting mixture were added **2** (0.20 mmol) and **4** (0.60 mmol). Then, the resulting mixture was degassed

(three times) under argon atmosphere. After that, TMSCN (0.60 mmol) was added into the mixture. At last, the mixture was stirred at a distance of ~1 cm from a 2 × 3 W purple LEDs at room temperature 6 h until the reaction was completed, as monitored by TLC analysis. The reaction mixture was diluted with water (10 mL). The mixture was firstly extracted with EtOAc (3 × 10 mL), then washed with NaHCO₃ (aq.; 15 mL), and finally washed with NaCl (aq.), dried over with anhydrous Na₂SO₄. After filtration and concentration, the residue was purified by silica gel chromatography afford the final products. Full experimental details and characterization of new compounds can be found in the Supplementary Methods.

**Preprint**. A previous version of this work was published as a preprint[69].

## Data availability

The authors declare that the main data supporting the findings of this study, including experimental procedures and compound characterization, are available within the article

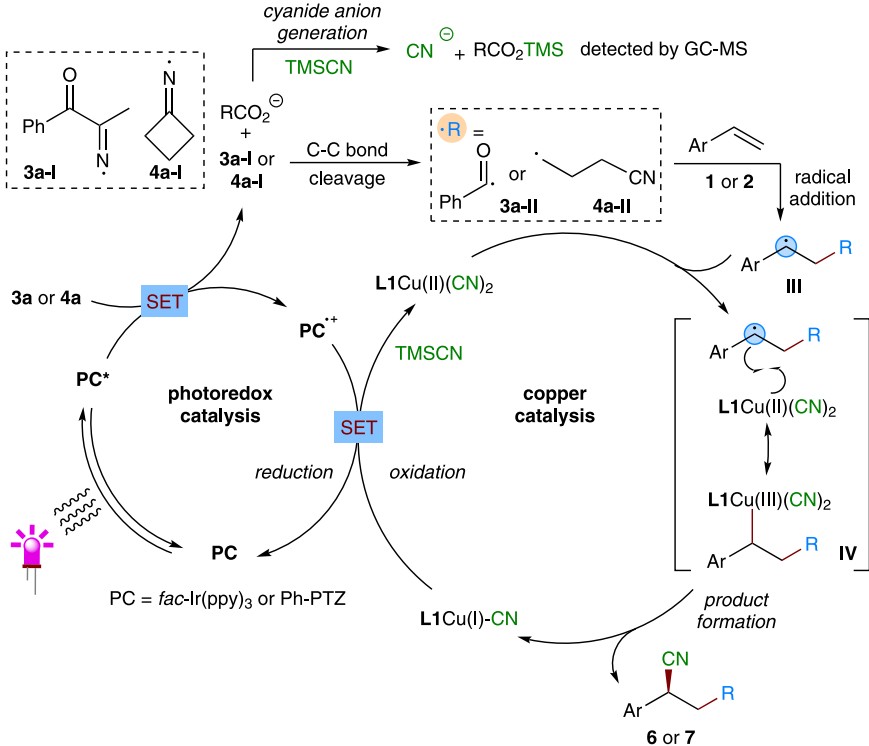

**Fig. 8 Proposed reaction mechanism.** SET single-electron transfer.

and its Supplementary Information files. X-ray structural data of compounds **6bj** and **7ja** are available free of charge from the Cambridge Crystallographic Data Center under the deposition number CCDC 2047031 (**6bj**) and 2047032 (**7ja**). These data can be obtained free of charge from The Cambridge Crystallographic Data Center via www.ccdc.cam.ac. uk/data_request/cif.

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

## Acknowledgements

We are grateful to the financial support from the National Natural Science Foundation of China (Nos. 21971081, 91856119, 21772053, 21820102003, and 91956201), the Program of Introducing Talents of Discipline to Universities of China (111 Program, B17019), and the Excellent Doctoral Dissertation Cultivation Grant to P.Z.W. from CCNU.

## Author contributions

P.-Z.W., Y.G., J.C. and X.-D.H. are responsible for the plan and implementation of the experimental work. J.-R.C. and W.-J.X. supervised the project and wrote the manuscript. All authors discussed the results and commented on the manuscript.

## Competing interests

The authors declare no competing interests.
