## [Peer Review File · Nature Communications]

REVIEWER COMMENTS

Reviewer #1 (Remarks to the Author):

Xiao, Chen and coauthors describe in this manuscript a novel catalytic protocol for the asymmetric dicarbofunctionalization of aryl alkenes. The reactions are enabled by the combination of radical addition with copper-mediated stereoselective radical-CN coupling, which is orchestrated in a highly efficient way. Visible light photocatalysis is employed to engender the acyl radicals and cyano radicals from oxime esters, and its catalytic cycle is merged nicely into recycling of the active copper species. This method allows a variety of β -cyano ketones and alkyldinitriles to be accessed in good yield and high enantioselectivity. It is impressive that only a tiny amount of catalysts is needed to guarantee a good result. The strategy revealed herein has general applicability for the construction of structurally useful optically active molecules from olefins. I think this work is worthy of publication in Nature Communications.

Other comments

In this study, purple LEDs were used as the light source to excite fac-Ir(ppy)₃ to its excited state. Had the authors tested other light sources such as blue LEDs and CFL lamps? Does the wavelength of the light have any notable effect on the result? If so, it should be commented in the manuscript.

Reviewer #2 (Remarks to the Author):

The manuscript reports an intermolecular enantioselective three-component radical dicarbofunctionalisation of alkenes using oxime esters as radical precursors and TMSCN. The chemistry requires the use of dual photoredox and copper catalysis. This is an interesting process, which however relies on established reactivity concepts.

The chemistry relies on the chemoselective addition of acyl and cyanoalkyl radicals, generated from the redox-active oxime esters by a photocatalytic N-centered iminyl radical-triggered C-C bond cleavage event, onto the alkenes to form new carbon radicals. Then the copper chiral catalyst intercepts the radical and trigger the formation of a new stereogenic centre with TMSCN. The feasibility of these two individual steps has been already demonstrated. The authors, and others, have used an external photocatalyst to activate redox-active oxime derivatives and generate iminyl radicals, which further triggered the formation of cyanoalkyl and acyl radicals via a C-C bond cleavage process (Refs 44-54). The stereoselective step is based instead on the nice copper-catalyzed radical relay strategy extensively developed by Guosheng Liu. Specifically, the same chiral ligand/copper combination used here was used by Liu to develop an enantioselective intermolecular cyano-trifluoromethylation of alkenes via a radical dicarbo-functionalisation process (see J. Am. Chem. Soc. 2016, 138, 48, 15547). Here the same asymmetric system has been combined with a different radical generation strategy, but the stereoselective step is essentially the same. Although interesting, this approach is therefore conceptually incremental over previous studies.

From a synthetic perspective, the chiral products in Figures 2 and 3 can be obtained by catalytic enantioselective conjugate addition strategies of cyanide to enones, as for example reported by Shibasaki (see J. Am. Chem. Soc. 2008, 130, 19, 6072 among other systems).

Overall, the present study is interesting but lacks the marks of originality, general advances and

broad synthetic interest required for publication on a general chemistry journal. This study seems better suited for a more specialized, synthetically-oriented audience.

Reviewer #3 (Remarks to the Author):

Chen and Xiao reported a cooperative photocatalysis and copper catalysis for the development of asymmetric difunctionalization of styrenes under mild conditions. The reaction reported an exciting approach to generate acyl radical to collaborate with copper catalyzed asymmetric cyanation, where the reaction was initiated by a photocatalyzed N-O bond cleavage to give a imino-radical, and followed by a C-C bond cleavage. The generated acyl radical added to styrene rapidly to generate benzylic radical, which was enantioselectively trapped by Box/Cu(II) cyanide. As described in manuscript, the reaction exhibited good substrate scope, where heterocycles could be well tolerated. Various of enantiomeric-enriched nitrile products were obtained with good to excellent ee values. Overall, this is a nice study on the asymmetric radical cyanations, and this referee recommends to publication in Nature Communication with minor revisions.

Comments:

- (1) For the acyl radical precursor, how about the reactivity of substrates derived from cyclic diketones?
- (2) During the optimization reaction conditions, the observation undesired side products should be given the yields, which can provide more information to get insight on the reaction mechanism.
- (3) The related asymmetric radical cyanation with the cooperative photocatalysis and copper catalyst (JACS 2017, 139, 15632) should be mentioned and cited in the manuscript.

Reply to Reviewer 1

My colleagues and I thank this referee very much for his/her favorable comments and helpful suggestions!

- (1) In this study, purple LEDs were used as the light source to excite *fac*-Ir(ppy)₃ to its excited state. Had the authors tested other light sources such as blue LEDs and CFL lamps? Does the wavelength of the light have any notable effect on the result? If so, it should be commented in the manuscript.

Response: According to this valuable suggestion, we examined other commonly used light sources including blue LEDs (2x3 W, $\lambda_{\text{max}} = 460 \text{ nm}$) and CFL lamp (40 W) in the model reaction. Both reactions also worked to give the desired products with 90% ee, but in only moderate yields due to decreased conversion and formation of considerable amounts of byproducts. These results showed that the wavelength of the light has notable effect on the reaction efficiency. These results have been added in Table 1 (entries 12 and 13) (page 4).

Table 1 | Optimization of the reaction conditions.

Entry	x / y / z [mol%]	Solvent	sp-1/sp-2/sp-3 [%] ^a	Yield [%] ^a	ee [%] ^b
1	0.5 / 0.6 / 1.0	DMA (0.05 M)	6/4/4	41	88
2	0.5 / 0.6 / 1.0	DMF (0.05 M)	2/5/1	39	89
3	0.5 / 0.6 / 1.0	CH ₃ CN (0.05 M)	1/11/-	6	90
4	0.5 / 0.6 / 1.0	THF (0.05 M)	1/9/-	15	90
5	0.5 / 0.6 / 1.0	CH ₂ Cl ₂ (0.05 M)	1/14/-	9	90
6	0.5 / 0.6 / 0.8	DMA (0.05 M)	7/3/2	64	88
7 ^c	0.5 / 0.6 / 0.8	DMA (0.05 M)	6/7/3	51	85
8 ^d	0.5 / 0.6 / 0.8	DMA (0.05 M)	8/2/2	69	81
9 ^e	0.5 / 0.6 / 0.8	DMA (0.05 M)	10/1/6	16	68
10	1.5 / 2.25 / 0.8	DMA (0.05 M)	6/12/1	78	90
11	1.5 / 2.25 / 0.8	DMA (0.04 M)	3/3/2	88 (74)	90
12^f	1.5 / 2.25 / 0.8	DMA (0.04 M)	5/33/1	41	90
13^g	1.5 / 2.25 / 0.8	DMA (0.04 M)	6/31/2	39	90

Reaction conditions: **1a** (0.1 mmol), **3a** (0.3 mmol), TMSCN (0.3 mmol), Cu(CH₃CN)₄PF₆ (x mol%), ligand **L1** (y mol%), *fac*-Ir(ppy)₃ (z mol%), solvent (2.0-2.5 mL), 2x3 W purple LEDs ($\lambda_{\text{max}} = 390 \text{ nm}$), at room temperature.

^aYields were determined by GC analysis, with isolated yield in parentheses.

^bDetermined by HPLC analysis on a chiral stationary phase.

^cWith CuCl.

^dWith CuI.

^eWith Cu(OTf)₂.

^fUnder the irradiation of 2x3 W blue LEDs ($\lambda_{\text{max}} = 460 \text{ nm}$) at room temperature.

^gUnder the irradiation of 2x3 W blue LEDs ($\lambda_{\text{max}} = 460 \text{ nm}$) at room temperature.

Moreover, we also obtained UV–Vis spectra of the solution containing 3a, 1a, $\text{Cu}(\text{CH}_3\text{CN})_4\text{PF}_6$ and photocatalyst alone, equimolar mixtures of $\text{Cu}(\text{CH}_3\text{CN})_4\text{PF}_6$, L1 and photocatalyst, as well as the reaction mixture (Figure 2). It was found that 3a, 1a, $\text{Cu}(\text{CH}_3\text{CN})_4\text{PF}_6$, and $\text{Cu}(\text{CH}_3\text{CN})_4\text{PF}_6/\text{L1}$ did not show any very strong absorption bands around the visible region. In contrast, the UV–Vis spectra of photocatalyst, mixture of $\text{Cu}(\text{CH}_3\text{CN})_4\text{PF}_6/\text{L1}/\text{PC}$ and reaction mixture have the same absorption band at 375 nm, which implied that photocatalyst *fac*- $\text{Ir}(\text{ppy})_3$ could be more efficiently excited by purple LEDs ($\lambda_{\text{max}} = 390 \text{ nm}$) than by blue LEDs ($\lambda_{\text{max}} = 460 \text{ nm}$). These data and the discussions have also been added in the revised manuscript (page 5).

Figure 2 | Absorption spectra in DMA of: (A) 3a; (B) 1a; (C) $\text{Cu}(\text{CH}_3\text{CN})_4\text{PF}_6$; (D) $\text{Cu}(\text{CH}_3\text{CN})_4\text{PF}_6/\text{L1}$; (E) *fac*- $\text{Ir}(\text{ppy})_3$; (F) $\text{Cu}(\text{CH}_3\text{CN})_4\text{PF}_6/\text{L1}/\text{fac}$ - $\text{Ir}(\text{ppy})_3$; (G) reaction mixture.

Reply to Reviewer 2

My colleagues and I thank this referee very much for his/her helpful suggestions!

(1) From a synthetic perspective, the chiral products in Figures 2 and 3 can be obtained by catalytic enantioselective conjugate addition strategies of cyanide to enones, as for example reported by Shibasaki (see J. Am. Chem. Soc. 2008, 130, 19, 6072 among other systems).

Response: I do appreciate this suggestion and agree with this reviewer's comments. The chiral products in Figures 2 and 3 can be obtained by catalytic enantioselective conjugate addition strategies of cyanide to enones, as for example reported by Shibasaki and others. In contrast to these known methods, our three-component protocol provides a new modular and generally applicable approach for the synthesis of structurally diverse chiral β -cyano ketones. Notably, the products in Figure 4 cannot be easily obtained by the previous strategies. In all, our approach is complementary to the methodologies reported by Shibasaki and others. These discussions and related literature have also been added in the revised manuscript (pages 12 and 16).

Reply to Reviewer 3

My colleagues and I thank this referee very much for his/her favorable comments and helpful suggestions!

(1) For the acyl radical precursor, how about the reactivity of substrates derived from cyclic diketones?

Response: According to this valuable suggestion, we preliminarily examined the cyclic oxime ester **3q** by reacting with **1a** and **5** with slightly increased amounts of catalysts. Unfortunately, the reaction proceeded slowly, giving only trace amount of desired product **6aq**, together with some byproducts **sp-1'**, **sp-2'**, and **sp-3'**. All further attempts met failure. These results and discussions have also been added in Figure 4 (page 7).

(2) During the optimization reaction conditions, the observation undesired side products should be given the yields, which can provide more information to get insight on the reaction mechanism.

Response: According to this valuable suggestion, the GC yields of the side products under various conditions have been added in Table 1 (page 4).

Table 1 | Optimization of the reaction conditions.

Entry	x / y / z [mol%]	Solvent	sp-1/sp-2/sp-3 [%] ^a	Yield [%] ^a	ee [%] ^b
1	0.5 / 0.6 / 1.0	DMA (0.05 M)	6/4/4	41	88
2	0.5 / 0.6 / 1.0	DMF (0.05 M)	2/5/1	39	89
3	0.5 / 0.6 / 1.0	CH ₃ CN (0.05 M)	1/11/-	6	90
4	0.5 / 0.6 / 1.0	THF (0.05 M)	1/9/-	15	90
5	0.5 / 0.6 / 1.0	CH ₂ Cl ₂ (0.05 M)	1/14/-	9	90
6	0.5 / 0.6 / 0.8	DMA (0.05 M)	7/3/2	64	88
7 ^c	0.5 / 0.6 / 0.8	DMA (0.05 M)	6/7/3	51	85
8 ^d	0.5 / 0.6 / 0.8	DMA (0.05 M)	8/2/2	69	81
9 ^e	0.5 / 0.6 / 0.8	DMA (0.05 M)	10/1/6	16	68
10	1.5 / 2.25 / 0.8	DMA (0.05 M)	6/12/1	78	90
11	1.5 / 2.25 / 0.8	DMA (0.04 M)	3/3/2	88 (74)	90
12^f	1.5 / 2.25 / 0.8	DMA (0.04 M)	5/33/1	41	90
13^g	1.5 / 2.25 / 0.8	DMA (0.04 M)	6/31/2	39	90

Reaction conditions: **1a** (0.1 mmol), **3a** (0.3 mmol), TMSCN (0.3 mmol), Cu(CH₃CN)₄PF₆ (x mol%), ligand **L1** (y mol%), *fac*-Ir(ppy)₃ (z mol%), solvent (2.0-2.5 mL), 2x3 W purple LEDs ($\lambda_{\text{max}} = 390 \text{ nm}$), at room temperature.

^aYields were determined by GC analysis, with isolated yield in parentheses.

^bDetermined by HPLC analysis on a chiral stationary phase.

^cWith CuCl.

^dWith CuI.

^eWith Cu(OTf)₂.

^fUnder the irradiation of 2x3 W blue LEDs ($\lambda_{\text{max}} = 460 \text{ nm}$) at room temperature.

^gUnder the irradiation of CFL (40 W) at room temperature. CFL = compact fluorescent lamp

ppy = 2-phenylpyridine. DMA = N,N-dimethylacetamide. DMF = N,N-dimethylformamide.

(3) The related asymmetric radical cyanation with the cooperative photocatalysis and copper catalyst (JACS 2017, 139, 15632) should be mentioned and cited in the manuscript.

Response: According to this suggestion, this paper has been incorporated as ref. 22 in the revised manuscript.

REVIEWERS' COMMENTS

Reviewer #1 (Remarks to the Author):

The Authors have revised the manuscript properly according to the reviewers' comments. I recommend it for publication.

Reviewer #3 (Remarks to the Author):

Author have well answered all the questions raised by this referee. Thus, this manuscript was recommended to publish in Nat. Commun.